# Dynamically expressed single ELAV/Hu orthologue *elavl2* of bees is required for learning and memory

Pinar Ustaoglu[1,2,6], Jatinder Kaur Gill[1,6], Nicolas Doubovetzky[3], Irmgard U. Haussmann[1,4], Thomas C. Dix[1,2], Roland Arnold [2,5], Jean-Marc Devaud[3] & Matthias Soller [1,5✉]

Changes in gene expression are a hallmark of learning and memory consolidation. Little is known about how alternative mRNA processing, particularly abundant in neuron-specific genes, contributes to these processes. Prototype RNA binding proteins of the neuronally expressed ELAV/Hu family are candidates for roles in learning and memory, but their capacity to cross-regulate and take over each other's functions complicate substantiation of such links. Honey bees *Apis mellifera* have only one *elav/Hu* family gene *elavl2*, that has functionally diversified by increasing alternative splicing including an evolutionary conserved microexon. RNAi knockdown demonstrates that ELAVL2 is required for learning and memory in bees. ELAVL2 is dynamically expressed with altered alternative splicing and subcellular localization in mushroom bodies, but not in other brain regions. Expression and alternative splicing of *elavl2* change during memory consolidation illustrating an alternative mRNA processing program as part of a local gene expression response underlying memory consolidation.

[1] School of Biosciences, College of Life and Environmental Sciences, University of Birmingham, Edgbaston, Birmingham B15 2TT, UK. [2] Birmingham Centre for Genome Biology, University of Birmingham, Edgbaston, Birmingham B15 2TT, UK. [3] Research Center on Animal Cognition (CRCA), Center for Integrative Biology (CBI), Toulouse University, CNRS, UPS, Toulouse 31062, France. [4] Department of Life Science, Faculty of Health, Education and Life Sciences, Birmingham City University, Birmingham B15 3TN, UK. [5] Institute of Cancer and Genomics Sciences, College of Medical and Dental Sciences, University of Birmingham, Edgbaston, Birmingham B15 2TT, UK. [6] These authors contributed equally: Pinar Ustaoglu, Jatinder Kaur Gill. ✉email: m.soller@bham.ac.uk

Changes in gene expression play pivotal roles in memory consolidation, the process through which memories are stabilized and stored into long-term memory[1–4]. A common feature of neuronal genes, particularly ion channel and cell adhesion genes, is their often complex pattern of alternative splicing, which alters protein coding and regulatory potential in flanking untranslated regions of the mRNA[5–7]. Alternative splicing events particularly in cell adhesion and ion channels among other genes have been linked to learning and memory[8–12], but little is known how RNA-binding proteins impact on alternative splicing programs that operate in learning and memory. Here, we focused on ELAV (Embryonic Lethal Abnormal Visual system)/Hu family RNA-binding proteins because they are prominently expressed in neurons of all metazoans, regulate alternative splicing and expression of synaptic genes as well as formation of new connections[13–17].

Like many RNA-binding proteins (RBPs) ELAV/Hu proteins comprise a family of highly related proteins in humans and many animals. Humans have four *ELAV/Hu* genes (*HuB, HuC, HuD,* and *HuR*), *Drosophila* have three (*elav, fne,* and *Rbp9*) and *Hydra* have three (*ELAV-like 1–3*)[18,19]. Some animals including the lancelet *B. floridae*, the nematode *C. elegans*, the honeybee *A. mellifera* and the cricket *G. bimaculatus*, however, have only one gene of an *ELAV/Hu* family orthologue indicating a very dynamic protein family with gains and losses during animal evolution[18,19]. Of note, the single *ELAV/Hu* family orthologue *found in neurons* in crickets has substantially expanded its coding capacity by alternative splicing to encode 24 protein isoforms[19].

In mice, all Hu proteins are expressed in largely overlapping patterns in mature neurons[20], while in *Drosophila* pan-neural expression of ELAV and FNE starts with the birth of neurons, and RBP9 is first detected in late larval neurons[21–24]. Although ELAV family RBPs in *Drosophila* have distinct neuronal phenotypes based on the analysis of null mutants and genetic interactions among them, they can cross regulate each other's targets depending on cellular localization and concentrations complicating the analysis of their functions[24].

ELAV/Hu proteins are prototype RBPs, which harbor three highly conserved RNA Recognition Motifs (RRMs). The first two RRMs are arranged in tandem and the third RRM is separated by a less-conserved hinge region. ELAV/Hu family RBPs bind to short, uridine-rich motifs, which are ubiquitously found in introns and untranslated regions, but ELAV/Hu proteins are gene-specific and have a complement of dedicated target genes[15,17,25–28]. Due to the prominent nuclear localization, ELAV in *Drosophila* has mostly been associated with gene-specific regulation of alternative splicing and polyadenylation, but it can also regulate mRNA stability[29–36]. Although the three RRMs comprise the evolutionary most conserved parts of ELAV/Hu proteins, individual members are to a large degree functionally interchangeable when adjusting expression levels and subcellular localization[24,37,38]. Hence, regulation of the activity of ELAV/Hu proteins likely occurs at the level of post-translational modifications and suggest that less conserved and unstructured linker sequences between or within RRMs serve fundamental functional roles, possibly by regulating interactions with other proteins[39].

To avoid complications of assigning specific gene functions to individual members of the ELAV/Hu family, we focused on honeybees whose genome encodes only one copy of an *elav/Hu* family gene[18], *elavl2*, an orthologue of *Drosophila fne*[22]. Conveniently, honeybees are a well-established model for the study of learning and memory. Here we show that the single *elavl2* gene in honeybees is required for learning, as well as the formation of stable memories by RNAi knockdown. Although bees have only a single *elav/Hu* family gene *elavl2*, its coding capacity proliferated by increasing alternative splicing to generate 40 different protein isoforms. The splicing pattern changes during development and between different adult social castes, but also shows variability among the brains of individual adult workers. Likewise, ELAVL2 expression changes in mushroom bodies (brain centers involved in learning and memory), but not in the medulla of the optic system, to generate individual expression patterns reminiscent of experience-dependent neuronal activity that forms the basis of gene expression changes associated with memory consolidation. Consistent with a role in learning and memory consolidation, *elavl2* expression and inclusion levels of alternative exons change during the early phases of memory consolidation. In this memory consolidation phase, also transcription is required and hence alternative splicing could be altered then depending on experience[40,41].

## Results

### ELAVL2 is required for learning and memory consolidation in bees after olfactory reward conditioning.

To detect bee ELAVL2, we used a polyclonal antiserum raised against *Drosophila* ELAV[42], that cross-reacts with bee ELAVL2 and human HuR, but not with other *Drosophila* ELAV family members and *Drosophila* cap methyltransferase CMTr1[43] as shown by Western blot from bacterially expressed GST-fusion proteins (Supplementary Fig. 1a, b).

The single ELAVL2 in bees is prominently expressed in the brain as determined by Western-blots recognizing the expected 38 kDa protein (Supplementary Fig. 1c). We did not detect ELAVL2 in bee muscle tissue, fat body, or gut (Supplementary Fig. 1c).

To assess whether ELAVL2 has a role in learning and memory in bees the single bee *elavl2* gene was knocked down by RNAi leading to a reduction of $76 \pm 5.1\%$ after two days ($n = 3$, Fig. 1a, Supplementary Fig. 1d, e). Two days after injection of *elavl2* or *GFP* control dsRNA, bees were individually trained and short-term memory was scored 2 h after training (Fig. 1b). Both groups showed significant learning over the successive trials (RM-ANOVA, *Trial* effect: $F = 61.93$, $p < 0.001$), but performance was affected by treatment (*Trial × Treatment* interaction: $F = 4.33$, $p < 0.05$). Indeed, as compared to controls, significantly fewer *elavl2* dsRNA-injected bees showed conditioned responses by the end of training (Fischer's test on 3rd trial: $\chi^2 = 4.22$, $p < 0.05$, Fig. 1c left). However, short-term memory retrieval remained unaffected ($\chi^2 = 0.64$, $p > 0.05$, Fig. 1c right).

We then asked whether *elavl2* knockdown might impact on the consolidation of long-term memory independently on its effect on acquisition. Therefore, injections and training were performed as before to ensure that *elavl2* levels would still be reduced during the hours following training (Fig. 1d), i.e. at a time when crucial transcriptional activity is required for long-term memory consolidation[40,41]. We then tested for their memory two days after training (a typical delay to assess consolidated long-term memory). In these conditions, learning occurred normally (RM-ANOVA, *Trial* effect: $F = 108.6$, $p < 0.001$; *Trial × Treatment* interaction: $F = 0.50$, $p > 0.05$; Fig. 1e left). Yet, the two groups showed different capacities to recall the memory of the CS-US association (Fischer's test: $\chi^2 = 10.08$, $p < 0.01$, Fig. 1e right). In addition, only control bees responded significantly more to the CS than to the novel odorant (*GFP*: $\chi^2 = 11.55$, $p < 0.001$; *elavl2*: $\chi^2 = 3.77$, $p > 0.05$).

To reject the possibility that loss of ELAVL2 impairs long-term memory retrieval per se due to a prolonged downregulation of *elavl2*, we performed an additional experiment in which injection was done shortly before training, when RNAi is not yet effective (Fig. 1f). As expected, this treatment did not affect learning (*Trial* effect: F = 62.93, p < 0.001; *Trial × Treatment* interaction:

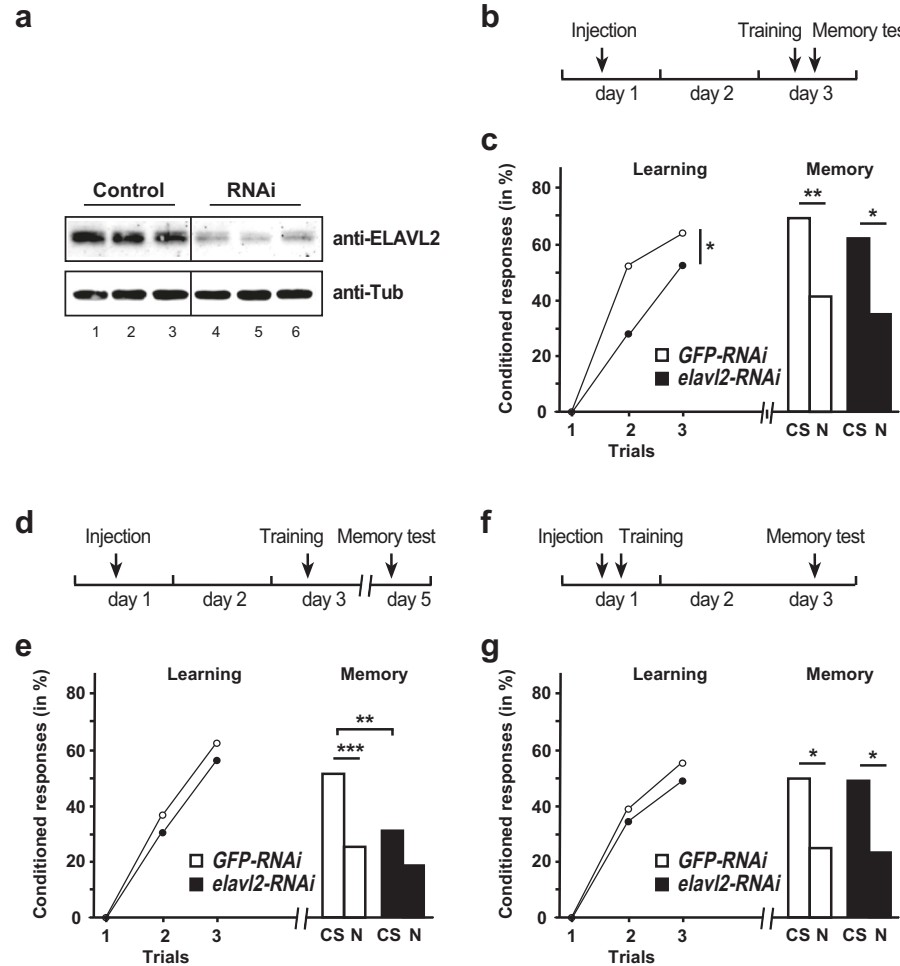

**Fig. 1 ELAVL2 is required for learning and memory consolidation. a** Western blot detecting ELAVL2 (top) or tubulin (bottom) in bee central brains of control *GFP* and *elavl2* dsRNA-injected workers 50 h after injection. **b** Schematic of the treatment to test for ELAVL2's role in learning. **c** Learning (*left*) and memory (*right*) performances of control *GFP* dsRNA (*white*, n = 66) and *elavl2* dsRNA (*black*, n = 74) injected worker bees. *CS*: conditioned odor, *N*: novel odorant. *$p < 0.05$; **$p < 0.01$. **d** Schematic of the treatment to test for ELAVL2's s role in memory consolidation. **e** Learning (*left*) and memory (*right*) performances of control *GFP* dsRNA (*white*, n = 74) and *elavl2* dsRNA (*black*, n = 77) injected worker bees. *CS*: conditioned stimulus, *N*: novel odorant **$p < 0.01$; ***$p < 0.001$. **f** Schematic of the treatment to test for ELAVL2's role in memory retrieval. **g** Learning (*left*) and memory (*right*) performances of control *GFP* dsRNA (*white*, n = 53) and *elavl2* dsRNA (*black*, n = 50) injected worker bees. *CS*: conditioned stimulus, *N*: novel odorant *: $p < 0.05$. The source data underlying this figure are available in Supplementary Data 1.

F = 0.15, $p > 0.05$; Fig. 1g left). More importantly, memory retrieval was intact and two days after training both groups responded similarly to the CS (Fischer's test: $\chi^2 = 0.02$, $p > 0.05$) and responded significantly less to the novel odorant (GFP: $\chi^2 = 6.24$, $p < 0.05$; *elavl2*: $\chi^2 = 5.66$, $p < 0.05$), thus indicating a preserved memory of the CS-US association.

These results thus argue that *elavl2* is required for the early formation of an associative memory over repeated acquisition trials, and for its subsequent consolidation.

**The single bee ELAVL2 gene is dynamically alternatively spliced.** The bee ELAVL2 protein is highly homologous to those of the *Drosophila* ELAV family (ELAV, FNE, and RBP9) in the three RRM domains, but diverges significantly in the unstructured hinge domain separating RRM2 from RRM3 (Fig. 2a and Supplementary Fig. 2). Given the much more sophisticated tasks associated with the social life of bees compared to *Drosophila*, it is surprising that bees have only one *elav/Hu* family gene[18,19]. However, diversification of gene function can also be achieved by increasing alternative splicing[5]. This prompted us to investigate whether *elavl2* in bees is alternatively spliced. Indeed, cloning of

full-length *elavl2* from RT-PCR revealed five alternatively spliced exons: exons 3a, exon 4a adding an additional 3'ss, exon 4b adding an additional 5'ss, exon 4c, and exon 4d, (Fig. 2a–c, Supplementary Fig. 2). The 45 nt exon 4c has been described as evolutionary conserved in insects[19,37,38]. The combination of these exons in addition to skipping of exon 4 variables potentially generates 40 different protein isoforms (Fig. 2a, Supplementary Data 1)[44].

Intriguingly, two of these alternative exons are located in the loop region of RRM2 and the other three are located in the hinge region (Fig. 2a–c, Supplementary Fig. 3a). Exon 4d is only 3 nts long and codes for a serine which can potentially be phosphorylated to impose further control of ELAV function[39]. Since the sequence of exon 4d is TAG and flanked by AG/GT consensus splice sites it is not a substrate for recursive splicing[45–47]. Exon 4d is too small to accommodate spliceosomal complexes on both sides and must thus be spliced sequentially[5].

Rather unexpectedly, we also detected splice variants, which skip exon 4 and its variants encoding the second half of RRM2 to result in proteins of 19–22 kDa. This results in truncated ELAVL2 proteins by introducing a frameshift removing much of the beta-sheet of RRM2 involved in RNA recognition as well as alpha-helix

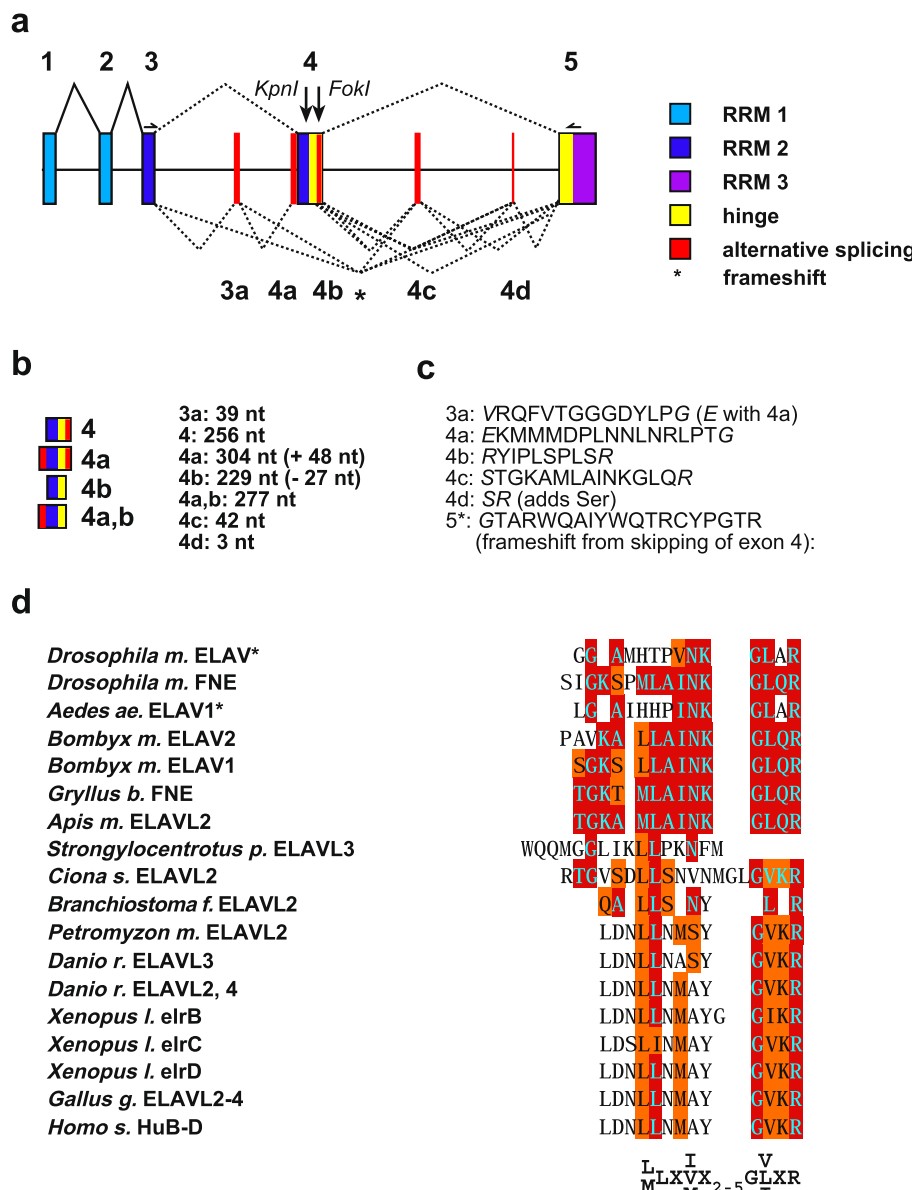

**Fig. 2 Single bee *elavl2* gene is alternatively spliced. a** Gene model of *Apis mellifera elavl2* gene depicting exons (boxes) and their splicing. Constant splicing is indicated by solid lines and alternative splicing is indicated by dashed lines. In total, 40 different alternative splice products are possible. Skipping of exon 4 variants results in a frameshift in exon 5 (indicated by an asterisk). **b** Depiction of exon 4 variants and length of alternative microexons. **c** Sequence of alternative microexons and frameshift in exon 5* from skipping of exon 4 variants. **d** Alignment of alternative microexon 4c from bees with a part present in *Drosophila* ELAV or alternatively spliced microexons in ELAV/Hu family proteins of other species with a consensus motif shown at the bottom. Retroposed genes lacking introns are indicated by an asterisk.

2 that makes up the backbone of the RRM structure. Since skipping of variable exons between the exon 3 and 4 deemed unfunctional based on RNA-binding assays[48], we employed molecular modeling to explore the capacity of frequently included alternative exons 3a and 4a to build alternative structures that might hold functionality. Inclusion of exon 3a with concomitant exclusion of exon 4, 4a and/or 4b adds an additional beta-sheet potentially increasing the capacity to bind RNA (Supplementary Fig. 3b). The inclusion of exon 4c further adds an additional alpha-helix likely stabilizing this alternative RRM structure (Supplementary Fig. 3c).

Intriguingly, exon 4c from bees has been found conserved in *Drosophila* FNE and aligns to part of ELAV[37,38,44]. Human ELAV/Hu family proteins also harbor an alternatively spliced small exon between the second and third RRM at a position

similar to that of the hinge region of bee *elavl2*[20,49], before a conserved motif involved in nuclear-cytoplasmic shuttling[50]. Alignment of exon 4c from bees with orthologues in other insects such as the mosquito *Aedes aegyptii*, the silk moth *Bombyx mori* and the cricket *Gryllus bimaculatus*[18,19], the sea urchin *Strongylocentrotus purpuratus*, the sea squirt *Ciona savignyi*, the lancelet *Branchiostoma lanceolatum*, the lamprey *Petromyzon marinus*, the zebrafish *Danio rerio*, the African clawed frog *Xenopus laevis*, the Chicken *Gallus gallus* and human ELAV/Hu proteins revealed a microexon at the same position with a consensus motif L/MLXI/V/MX$_{2-5}$GV/L/IXR (Fig. 2d), which is consistent with an evolutionary conserved microexon program between vertebrates and invertebrates[51,52].

Bees also have a 3 nt microexon (Fig. 2a–c), that adds a serine, which potentially can be phosphorylated[39]. In vertebrates, this

serine is added through an alternative 3'splice site at the same position in Human *HuB* and *HuC*, chicken *ELAVL2-4*, Xenopus *elrB*, and *elrC*, and zebrafish *ELAVL2* and *ELAVL3*.

Next, we analyzed alternative splicing in more detail than possible on agarose gels, where multiple alternative splice products, amplified from mRNA of larval brains were detected only as a smear (Fig. 3a). Therefore, we employed a higher resolution separation of $^{32}$P-labeled PCR products using denaturing polyacrylamide gels. This analysis revealed 23 distinguishable products with sizes between 78 and 463 nt (Fig. 3b, c). Most frequently found splice variants were 3-4-5, 3-3a-4-4c-5, 3-3a-4-5/3-4-4c-5 and 3-3a-4b-5/3-4b-4c-5 as well as the truncated isoforms 3-4c-5 and 3–5, thus indicating functional relevance for the newly identified alternative splice products.

Since some of the splice variants were not separable based on size, we wanted to determine how frequently each alternative exon is included. For this purpose, we digested 5′ $^{32}$P-labeled PCR products with KpnI or FokI restriction endonucleases to cleave off their unlabeled 3′ parts (Figs. 2a and 3d, e). For both sides of exon 4, all possible combinations of alternative splice products were detected.

Next, we analyzed the ELAVL2 alternative splicing pattern at different developmental stages and in different tissues ($n = 3$, Fig. 3f–h and Supplementary Fig. 4). This analysis revealed dynamic inclusion of alternative exons. In particular, splicing from exon 3 to 4 is absent in embryos compared to larval brains and adults (Fig. 3f left, $p \leq 0.001$). Skipping of exon 4, 4a or 4b leads to significantly increased abundance of the truncated isoforms 3-4c-5 and 3-3a-4c-5 in adults (Fig. 3f right, $p \leq 0.05$).

To obtain further insights into the dynamics of *elavl2* alternative exon use at a cellular level we performed whole-mount RNA in situ hybridization with antisense probes against alternative exons 3a and 4c in brains of worker bees (Fig. 4a–r, Supplementary Table 1). Most strikingly, both exons 3a and 4c show very dynamic inclusion levels in the mushroom bodies, displaying unique patterns in each individual bee (Fig. 4a, d, j, m, and see Supplementary Fig. 5a for a whole-brain image). In contrast, inclusion levels in the medulla (visual neuropil not involved in the learning process) are uniform for both splice variants (Fig. 4g, p). Control stainings with a probe against *Drosophila tango13* alternative exon 7b did not stain and a probe against constant exon 14 of *Apis Dscam* stained ubiquitously in the mushroom bodies (Supplementary Fig. 5b–g). Likewise, a probe against *Drosophila elav* only stained D*rosophila* embryos, but not mushroom bodies of bees and vice versa, a probe against bee *elavl2* exon 4c only stained bee mushroom bodies, but not *Drosophila* embryos (Supplementary Fig. 5h–k).

**ELAVL2 protein levels are dynamic in mushroom bodies of worker bees**. ELAV family proteins are pan-neurally expressed in *Drosophila*. Their expression seems not to be dynamic as judged from antibody stainings, but changes in nuclear and cytoplasmic distributions have been observed[24]. When we then analysed ELAVL2 expression in mushroom bodies, we found that expression varied between individual bees (Fig. 5a–i, Supplementary Fig. 5a). In some bees, ELAVL2 localizes to the nucleus (Fig. 5c–e) while in others it was cytoplasmic (Fig. 5f–h), or both nuclear and cytoplasmic (Fig. 5i–k). In addition, we detected areas where ELAVL2 was absent (Fig. 5f–h) or levels were reduced (Fig. 5i–k). Quantification of ELAVL2's cellular localization revealed that ELAVL2 is nuclear in about 75% of worker bee brains (Fig. 5c–e, l). In the remaining 25%, however, ELAVL2 expression was very dynamic, showing patches of nuclear and cytoplasmic localization, but also small patches of cells with no ELAVL2 expression (Fig. 5f–k, l). Because these analyses were

done on animals whose previous experience in the field could not be controlled, we wondered whether such localized changes in ELAVL2 expression might be indicative of experience-dependent plasticity.

**ELAVL2 expression and alternative splicing is altered upon learning**. Since bees depend on learning and memory to forage, the pronounced loss of ELAVL2 expression in some of the brains of worker bees might reflect interindividual learning/memory variations. Thus, we thought of testing if such local down-regulation might be indicative of a particular individual learning/memory status. To increase the sensitivity of our follow-up molecular analysis we took advantage of the diversity in the speed of learning observed among individuals during a 5-trial training by splitting trained bees into fast and slow learners, e.g., bees that responded in the first two trials and every time after the initial response vs bees with a lack of response in the first two trials or with gaps after the initial response (Fig. 6a). We then monitored *elavl2* expression levels from their brains by qPCR at various timepoints after training (Fig. 6b). This analysis indeed revealed that *elavl2* steady-state mRNA levels had dropped 50% two hours after training in the fast learners compared to slow learners. We therefore thought to analyze alternative splicing of *elavl2* exons 3a and 4c 2 h after training. We detected a significant increase in the inclusion of exons 3a and 4c in the mushroom bodies, but not in the medulla one hour after training in fast learners (Fig. 6c–e). We also analyzed the alternative splicing pattern of *elavl2* on denaturing polyacrylamide gels, but no differences were detected after learning in this assay, likely because the observed changes occurred only in relatively few cells (Supplementary Fig. 6).

## Discussion

Many RNA-binding proteins including neuronal ELAV/Hu RBPs are comprised of families of highly related proteins[15,53]. In case of ELAV family RBPs, they have unique individual functions, but depending on cellular localization and concentrations they can cross-regulate targets making the study of their individual functions difficult[24,37,38,48]. Therefore we took advantage of honeybees due to the presence of only a single *elav* gene to examine whether ELAVL2 is required for learning and memory.

**A role for ELAVL2 in learning and memory**. Although neuronal ELAV/Hu family proteins are broadly expressed in the brain, mutants of individual genes in mice and *Drosophila* revealed only subtle developmental defects thus pointing towards a primary role in regulating neuronal functions as e.g. operating in learning and memory[17,24,54,55]. A knock-out of HuC in mice revealed a role in the synthesis of the neurotransmitter glutamate resulting in reduced neuronal excitability and impaired motor function[17]. For HuD, roles in learning and memory have been suggested due to its involvement in regulating GAP43 expression which has established roles in learning and memory[56–58]. Here, over-expression of HuD, which is cytoplasmic, leads to increased GAP-43 expression by increasing mRNA stability. Since in bees *elavl2* steady-state mRNA levels drop for a short period after training early during memory consolidation, this might reflect functional compartmentalization of ELAV/Hu family proteins between nucleus and cytoplasm as bee ELAVL2 is mostly nuclear compared to HuD, which is mostly cytoplasmic in a learning context in the mouse hippocampus[57].

The changes in *elavl2* expression in the brain occur within two hours following learning consistent with a role in memory consolidation. Indeed, our learning protocol was designed to trigger the formation of stable long-term memories, which can be detected several days later[59]. Such memories are formed through

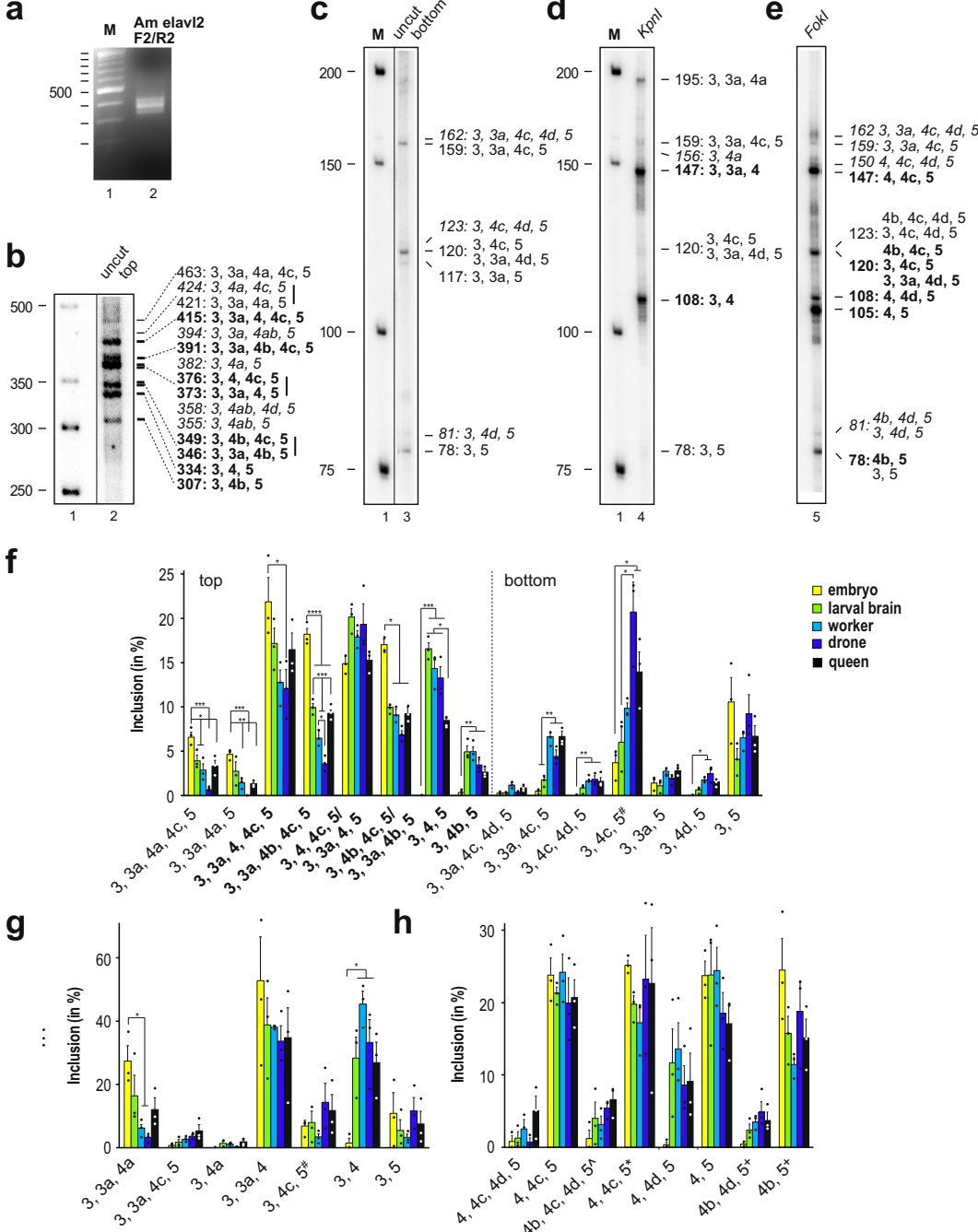

**Fig. 3 _elavl2_ alternative splicing is dynamic during development. a** Multiple products are detected from RT-PCR of RNA from larval brain in the alternatively spliced part of bee _elavl2_ on an agarose gel. M: Marker **b, c** Top and bottom part of a representative 5% denaturing polyacrylamide gel separating ³²P-labeled alternative splice products from larval brains. Length of PCR products from splice variants are indicated at the right in bold for prominent products and in italics for very rare products. Vertical lines at the right of indicated splice variants indicate inseparable products. M: Marker. **d, e** Analysis of alternative splicing from larval brains proximal (_Kpn_I) and distal (_Fok_I) of exon 4 from ³²P-labeled labeled forward (**d**) or return (**e**) primer after digestion with either _Kpn_I or _Fok_I on a representative 5% denaturing polyacrylamide gels. Length of PCR products from splice variants are indicated at the right in bold for prominent products and in italics for very rare products. M: Marker. **f–h** Developmental and sex-specific alternative splicing of bee _elavl2_ quantified from denaturing polyacrylamide gels shown as mean with the standard error from three independent replicates as percent from all splice products from top (**f** left) and bottom (**f** right) gel parts and after _Kpn_I or _Fok_I digestion as above from embryos (yellow), larval brains (green), drone brains (light blue), worker brains (dark blue) and queen brains (black). (**f** and **g**). #The 120 nt product can be either 3, 4c, 5 or 3, 3a, 4d, 5. (**h**) ^The 123 nt products are either 4b, 4c, 4d, 5 or 3, 4c, 4d, 5. *The 120 nt products are either 4b, 4c, 5 or 3, 4c, 5 or 3, 3a, 4d, 5. +The 80 nt products are either 4b, 5 or 3, 5 +/− 4d. The source data underlying this figure are available in Supplementary Data 1.

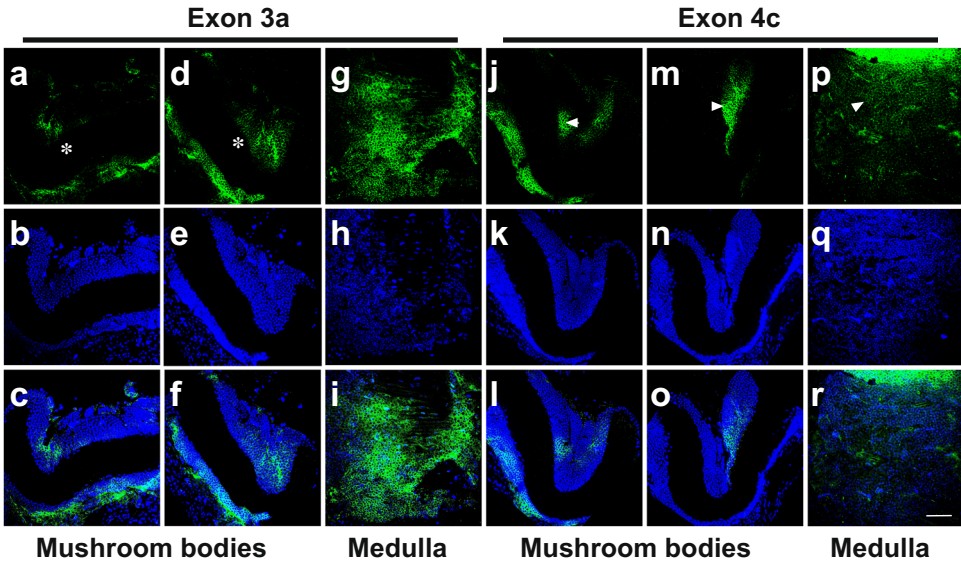

**Fig. 4 Alternative splicing of *elavl2* exons 3a and 4c is dynamic in mushroom bodies.** Representative RNA in situ hybridizations in worker bees against *elavl2* exon 3a (**a**, **d**, and **g**) and exon 4c (**j**, **m**, and **p**) in mushroom bodies (**a**, **d**, **j**, and **m**) and the medulla (**g** and **p**) counterstained with DAPI to visualize nuclei (**b**, **e**, **h**, **k**, **n**, and **q**) and merged pictures (**c**, **f**, **i**, **l**, **o**, and **r**). Scale bar in R is 30 μm.

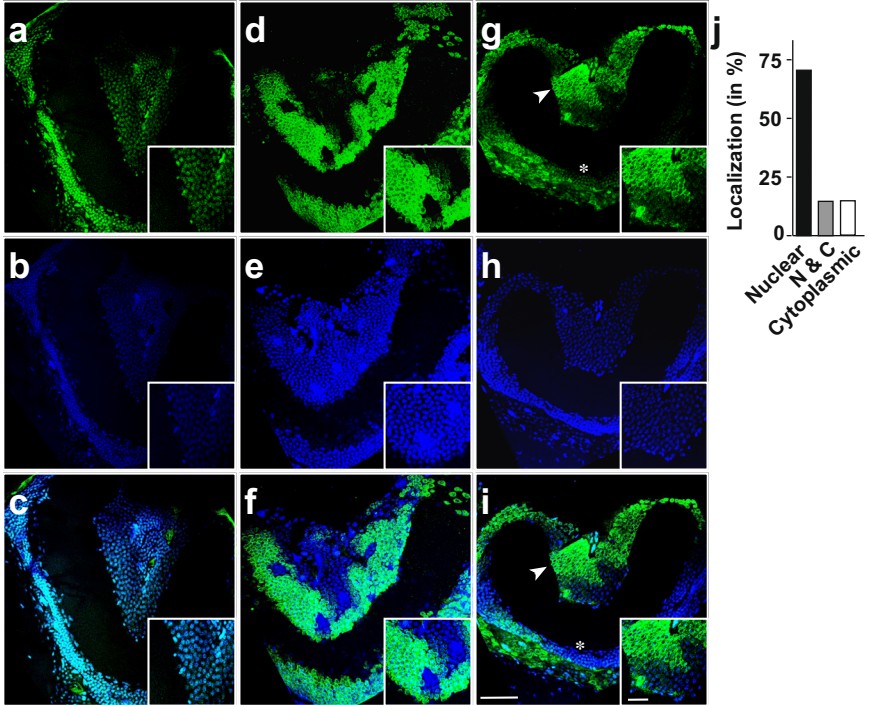

**Fig. 5 Localization and expression levels of ELAVL2 are dynamic in mushroom bodies.** Representative anti-ELAV antibody stainings in worker bees of mushbodies (**a**, **d**, and **g**) counterstained with DAPI to visualize nuclei (**b**, **e**, and **h**) and merged pictures (**c**, **f**, and **i**). Insets show higher magnifications of ubiquitous expression and nuclear localization (inset in **a**–**c**), patchy expression with mostly cytoplasmic localization (inset in **d**–**f**) and patchy expression with nuclear and cytoplasmic expression (inset in **g**–**i**, arrowhead in **g** and **i**). The asterisk in **g** and **i** indicates nuclear localization in the lower part of the mushroom body. A summary of ELAVL2 localization in mushroom bodies of worker bees is shown in panel **j** ($n = 20$). Scale bars are 30 μm in **i** and 6 μm in the inset. The source data underlying this figure are available in Supplementary Data 1.

a consolidation process initiated before the end of training and within a few hours, which depends on gene transcription[40,41]. It is therefore conceivable, that altered levels of ELAVL2 will impact on newly transcribed genes. In particular, expression of ELAV has been linked to implementing splicing programs governing neuronal characteristics such as changes in cell adhesion. Potentially, reduction of *elavl2* levels could reduce cell adhesion

for facilitating the creation or pruning of new synaptic connections. Indeed, changes in connectivity, particularly in the mushroom bodies, is an important process underlying long-term memory formation[60]. Such role is well in agreement with our observations in *Drosophila*, where reducing alternative splicing of the ELAV target *ewg*, a transcription factor, results in increased growth of synapses at the NMJ[16,61]. Likewise, we observed

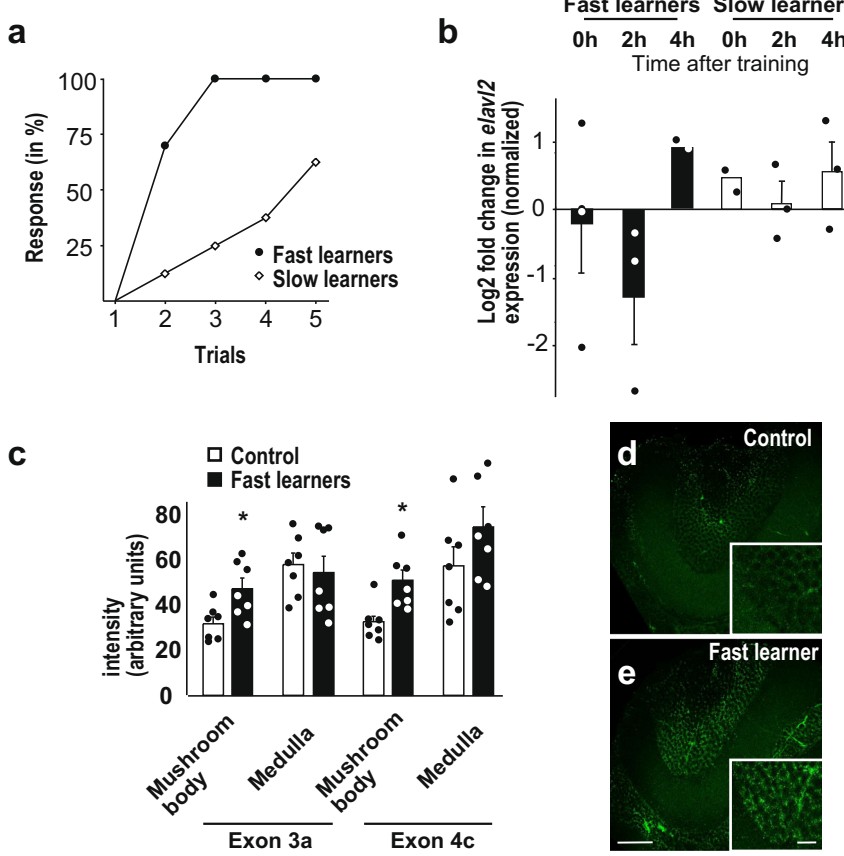

**Fig. 6 *elavl2* alternative splicing and expression levels change upon learning. a** Learning response of bees grouped into fast and slow learners (fast learners $n = 10$ and slow learners $n = 8$). **b** *elavl2* mRNA expression levels determined by qPCR 0, 2, and 4 h after training shown as mean with the standard error from two to four independent replicates central brains normalized to *Appl* expression ($p = 0.13$). **c** Inclusion levels of *elavl2* exons 3a and 4c quantified from RNA in situ hybridizations from control (unpaired) and fast learning (paired) bees in the mushroom body and the medulla ($p < 0.05$, $n = 7$). **d**, **e** Representative RNA in situ hybridizations against exon 3c in mushroom bodies of control (**d**) and fast learning (**e**) bees 2 h after training. Scale bars are 30 μm in E and 6 μm in the inset. The source data underlying this figure are available in Supplementary Data 1.

changes in ELAVL2 alternative splicing in bees leading to an increase in exon 3a and 4c inclusion, which is anticipated to have profound effects on target mRNA binding. In addition, skipping of exon 4, 4a, or 4b leads to a frameshift and an altered structure of the third RRM, which will alter target specificity and/or reduce binding affinity.

**Local and dynamic expression changes of ELAVL2 as a hallmark for its role in learning and memory.** A hallmark of memory formation is altered local gene expression followed by local changes of neuronal properties and establishment of new connections[1,3]. Activity-induced expression of immediate-early gene transcription factors has been associated with memory[40,62–67]. Intriguingly, expression of HuD can be induced by neuronal activity[68]. Stabilization of C/EBP by apELAV1 in *Aplysia* accompanies long-term memory[69], although apELAV1 is mainly nuclear in contrast to apELAV2, which is also cytoplasmic[70].

In agreement with a role for ELAVL2 in memory formation we find variable expression patterns for ELAVL2 in the mushroom bodies of worker bees. Even more compelling, the expression pattern of ELAVL2 in the mushroom bodies of worker bees is unique and differs between individuals. Similarly, inclusion levels of alternative exons 3a and 4c also show unique patterns in each individual bee. This can be understood as possible consequences of differences in the previous experience that individuals have had, either within the hive or outdoors (e.g., social interactions,

environmental stimuli). The rapid changes of ELAVL2 expression observed after learning occur in the same time-frame as activity-induced expression of immediate-early genes[71] suggesting a role for alternative splicing during the early phases of memory consolidation, which requires transcription in addition to protein synthesis from stored mRNA at synapses[4,40,41]. Notably, *elav* expression in *Drosophila* is controlled by miRNAs that can contribute to fast changes in expression, but also restrict the expression of ELAV protein to the nervous system[72]. Whether ELAV family proteins are expressed in other cells than neurons, as found for human HuR and suggested from *fne* mRNA expression in crickets, remains to be determined[19].

Mushroom body connectivity is shaped by individual experience during a continuous maturation process[73,74]. Yet, molecular tools available in *Drosophila* or mice to genetically label individual neurons are currently lacking in the honeybee in order to identify those neurons where *elavl2* expression varies and to establish interindividual variations[75]. In addition, ELAVL2's cellular localization also varied in individual cells in the mushroom bodies from nuclear to cytoplasmic. Such differences in cellular localization are expected since *Drosophila* ELAV localizes mostly to the nucleus, RBP9 is cytoplasmic and FNE is found in both compartments. However, ELAV/Hu family proteins also shuttle between nucleus and cytoplasm[24,76]. Upon removal of ELAV in *Drosophila*, alternatively spliced microexon 4c is included in FNE leading to nuclear localization and regulation of alternative splicing of genes that are otherwise

ELAV targets[37,38] suggesting a complex network of interactions among ELAV/Hu proteins.

Alternative splicing could serve as an adaptive mechanism to changes in perception, but also to environmental conditions such as toxic insult[77]. Although learning and memory is affected by neonicotinoids in insects, we did not find any changes in *elavl2* alternative splicing[44,78,79]. Since we also could not detect any alternative splicing changes after learning in mRNA from central brains, drastic changes in alternative splicing relevant to learning and memory might occur only in few cells.

### Alternative splicing of a microexon in ELAV/Hu proteins is evolutionary ancient.

Human HuB-D genes contain an alternatively spliced microexon in the hinge region between the second and third RRM[20,80,81]. We identified an alternatively spliced exon at the same position in the single bee *elavl2* gene[44] homologous to a previously identified alternative exon in a cricket[19]. Comparison of the sequence between human and insects of this exon shows a high sequence similarity indicating that this exon is evolutionary ancient. Although we identified a consensus motif in exon 4c, convergent evolution is also possible due to the short sequence[51]. Previously, exon duplication between humans and *Drosophila* has been documented in a few ion channel genes leading to alternatively spliced exons at the same position, but without sequence conservation and no longer exons have been found that are evolutionary conserved between invertebrates and vertebrates[51,52]. Intriguingly, ELAV in *Drosophila* has lost its introns due to retrotransposition, but retained microexon 4c[18]. This microexon is involved in regulating nuclear localization of ELAV and FNE in Drosophila[37,38,50], and also affects localization of HuD in human cells[80,81]. Its increased expression shortly after training thus coincides with an initial nuclear role of ELAVL2 at the memory consolidation phase, which requires transcription[40,41]. The unstructured hinge region between the RRMs 2 and 3 in ELAV/Hu family RBPs has expanded by alternative splicing in humans[80], some insects[18,19] and in the sea squirt *Ciona savignyi* with unexpected complexity of eight alternative exons in this species (www.ensembl.org, ENSCSAVG00000003440), but the functional consequences of alternative splicing in this part of ELAV/Hu family proteins have not been determined.

For most neuronally alternative spliced microexons in mice, Srrm4 is required for their inclusion[82]. Srrm4 contains a novel evolutionary conserved protein domain 'enhancer of microexons' (eMIC) that is present in *Drosophila* Srrm2/3/4 and required for exon inclusion in the *Dscam* exon 9 cluster[83] indicating a conserved neuronal microexon program is present in vertebrates and insects[52].

### Alternative splicing in bee ELAVL2 is confined to unstructured linker regions, but not RNA recognition domains.

A main question arising from the presence of multiple highly related genes is whether they act in an overlapping manner. In case of *Drosophila* ELAV family members ELAV, FNE, and RBP9, distinct mutant phenotypes and the lack of major genetic interactions among them suggests largely independent functions[24]. However, cross-regulation between FNE and RBP9 is present in the regulation of synapse numbers. Likewise, expression in non-neuronal cells or swapping of expression and localization regulatory regions can to a large degree substitute for their individual functions and they can cross-regulate. Overlapping functions even extend to more distantly related Sex lethal (Sxl), which is required for neuronal functions in *Diptera*, but has been recruited in *Drosophila* for sex determination and dosage compensation[84]. Here, RBP9 is required for maternal inhibition of dosage compensation, a function that is taken over entirely by Sxl during

embryogenesis[24]. In addition, the ELAV binding site in *Drosophila virilis ewg* diverged substantially and does not align to the *D. melanogaster* ELAV binding site, but ELAV regulation is maintained[28].

These facts point out that the main distinction among ELAV family members only minimally occurs at the level of RNA recognition. Hence, it is conceivable, that the ELAV family in bees has "merged back" into a single copy gene by incorporating the variable parts between family members by alternative splicing as observed in the honeybee and cricket[19]. In this respect, it is very interesting that alternative splicing in bee ELAVL2 and cricket FNE occurs in unstructured linker regions between RRMs[19]. It is conceivable, that these regions mediate protein-protein interactions leading to sub-functionalization. Accordingly, the microexon present in the hinge region likely serves such purpose, but the interacting proteins remain to be identified.

Mis-regulation of microexons has been found as a major cause of autism spectrum disorders revealing essential functions for such microexons in neurons[82]. Notably, inclusion levels of this microexon in bees is altered upon learning and memory formation. Hence, lack of dynamic inclusion of microexons in ELAV/Hu family proteins might point toward a role in establishing the extensive memories often associated with some autism spectrum disorders[85].

## Materials and Methods

**Honeybees and treatment.** Honeybees (*Apis mellifera*) were collected from flowers or local bee hives in the UK for molecular biology experiments (worker bees were used unless otherwise specified). For behavioral experiments, workers were taken from the experimental apiary on the university campus in Toulouse (France), on the morning of each experiment. Following cold-anesthesia, they were harnessed in metal tubes leaving access to the head, fed with 5 µl of sucrose solution (50% weight/weight in water) and then kept in the dark at room temperature until needed. They were fed in the same way on every morning and evening during the time of each experiment.

**Behavioral assays.** Learning and memory capacities were assessed using a standard protocol based on the olfactory conditioning of the proboscis extension response (PER)[86], which consisted of three learning trials (unless specified otherwise) where animals were trained individually to associate an odorant with a sucrose reward as detailed below. Memory of the association was tested either one hour (short-term memory) or 48 h (long-term memory) after the last learning trial. In all experiments, bees of both treatment groups were trained in parallel. Each learning trial (40 s) started when the restrained bee was placed in front of an odorless airflow. After 15 s, the setup allowed to deliver an odor (conditioned stimulus, CS) for 4 s by partially diverting the flow in a syringe containing a filter paper soaked with 4 µl of pure odorant. (1-hexanol and 1-nonanol were used, alternatively for different bees; data were pooled after checking for any significant effect of the odorant used). Sucrose (unconditioned stimulus, US: same solution as for feeding) was delivered to the antennae using a toothpick, 3 s after CS onset, for 3 s. This triggered the bee's reflex extension of the proboscis to lick the reward. Whenever the animal already responded to the CS (conditioned response), it was directly allowed to feed upon US onset. Successive learning trials were separated by 10-min intervals to facilitate memory consolidation[59]. Memory was assessed by placing the animals again in the conditioning setup, and by presenting them the CS without the US[86]. The presence or absence of a conditioned response was recorded. In case of no response, sucrose was applied to the antennae at the end of the test, to control for an intact motor response. Bees failing to show an intact reflex were discarded. Bees that responded to the training in the first two trials and that responded every time were classified as fast learners. Bees that responded only two times in the four trials were classified as slow learners. The sucrose and odorants were purchased from Sigma-Aldrich (France).

**Recombinant DNA technology, RT-PCR, qPCR, and analysis of alternative splicing.** The sequence of oligonucleotides used in this study is listed in Supplementary Table 1. Recombinant DNA technology was done according to standard procedures as described[29]. Bee *elavl2* was amplified from oligo dT primed cDNA made from larval brains using primers AM elav F1 and AM elav R1 and cloned into a modified pBS SK+ using NgoMIV and XbaI. Clones (n = 45) were sequenced using primers elav F1 and elav R1.

RNA extraction from whole bees or dissected bee brains and RT-PCR was done as described[87]. Expression of *elavl2* at different timepoints was compared to *Appl* expression using primers AM elav qF3 and AM elav qR3 to amplify the constant

part of *elav* and normalized to unpaired control animals using qPCR as described[33,79].

For high-resolution analysis of *elavl2* alternative splicing primers AM elav F2 and AM elav R2 were used to amplify *elavl2* from cDNA. One of the primers was labeled using gamma[32]P-ATP (NEN) and PCR products were separated on sequencing type denaturing polyacrylamide gels. Polyacrylamide gels were dried, exposed to phosphoimager screens (BioRad), and quantified with QuantityOne (BioRad).

**RNAi, Western analysis recombinant protein expression, RNA in situ, antibody staining, and imaging**. For RNAi knockdown in bees, *elavl2* and GFP DNA templates for in vitro transcription were amplified for *elavl2* from a cloned cDNA with primers AM ELAV T7 RNAi F1 and AM ELAV T7 RNAi R1 and for *GFP* a 700 bp fragment was amplified with primers GFP T7 RNAi F1 and GFP T7 RNAi R1. Double-stranded RNA was generated by in vitro transcription with T7 polymerase with the MegaScript kit (Ambion) for 3 h according to the manufacturer's instructions. After digestion of the template with TurboDNAse (Ambion), dsRNA was phenol/chloroform extracted, ethanol precipitated and taken up in RNAse free water at a concentration of 5 µg/µl. The dsRNA (250 nl) was then injected into the brain through the median ocellus with a Nanoject II microinjector (Drummond).

RNAi efficiency testing for ELAVL2 was done from dissected central brains by Western blotting according to standard protocols as described[29] using a polyclonal rat anti-ELAV antibody generated against *Drosophila* ELAV (1:800)[42] and secondary HRP-coupled goat antirat antibody (1:5000, GE Healthcare) by chemiluminescence detection (Pierce) according to the manufacturer's instructions. Alternatively, infrared dye coupled secondary antibodies (IRDye800CW, LI-COR) were used and detected with an Odyssey infrared imaging system (LI-COR). A polyclonal antiserum raised in rabbits against *Drosophila* ELAV also cross-reacts with bee ELAVL2[38]. Tubulin was detected with a mouse anti-alpha tubulin antibody (1:10,000, clone DM1A, SIGMA). Quantification of Western blots was done with Quantity ONE 4.6.8 (BioRad) according to the manufacturer's instructions.

Recombinant bee ELAVL2 was made in *E.coli* by cloning the cDNA with primers GST AM ELAV F1 and primers GST AM ELAV R1 into a modified pGEX. *Drosophila* GST ELAV, GST FNE, GST RBP9, and human GST HuR, and GST dCMTr were as described[24,43]. For protein expression, 1.5 ml of a 3 ml overnight culture was diluted with 2 ml 2YT and IPTG (1 mM final) and proteins were induced for 8 h. For protein gels, 0.5 ml cells were pelleted and taken up in 50 µl 2x SDS loading buffer, boiled and 5 µl loaded onto an 8% SDS gel. For Westerns, proteins were further diluted 1:100.

Brain antibody stainings were done with rat polyclonal anti-ELAV antibody (1:200) and an antirat FITC labeled antibody (1:200, Molecular Probes) for two days each as described[16] and counterstained with DAPI (1 µg/ml).

To make probes for RNA in situ hybridizations, a pBS SK+ vector was modified by cloning a U-rich stem loop at the end of the in vitro transcript using EcoRI and KpnI and phosphorylated and annealed oligos RNA IS stem 1A and B, tango13A and B. ELAVL2 alternative exons were then cloned with XhoI and PstI using phosphorylated and annealed oligos AM elav 3a A and B, and AM elav 4c A and B. The sequence of RNA in situ probes used in this study are listed in Supplementary Table 1. Vectors were linearized with Acc56I and DIG-dUTP (Roche) labeled antisense transcripts were generated by in vitro transcription with T3 RNA polymerase for *Apis elavl2* exon 3a and 4c, and *Drosophila tango13* probes, and with T7 RNA polymerase for *Apis Dscam* exon 14 in 10 µl from 1 µg template DNA. After digestion of the template with TurboDNAse (Ambion), these transcripts were cleaned by centrifugation through a G50 Microspin column (GE Healthcare) in a final volume of 50 µl.

For in situ hybridizations, whole brains were fixed 30 min in 4% paraformaldehyde in PBT (PBS, 0.1% Tween 20) and then washed in PBT. Hybridizations were then done in 50% formamide buffer[88] (50% formamide, 5x SSPE, 50 µg/ml heparin (SIGMA), 0.1% Tween 20, 0.5 mg/ml denatured Salm sperm DNA) using 1:500 diluted probes at 39 °C for 3d as described[89]. To wash off unhybridized probe, tissues were incubated overnight in hybridization buffer at 39 °C. Brains were then washed with PBT and DIG-labeled probes were visualized by incubation with a sheep anti-DIG antibody (1:400, 2 days, Roche), and after washing followed by incubation with an FITC conjugated anti-sheep antibody (1:200, 2 days, SIGMA) and counterstained with DAPI (1 µg/ml). To evaluate that the probe concentration was adequate, bee brains or *Drosophila* embryos were incubated for two days in anti-DIG Fab fragments coupled to alkaline phosphatase (1:400) after hybridization and washing, and detected with NBT/BCIP (1:50, Roche) in TLMNT (100 mM Tris/HCl pH 9.5, 100 mM NaCl, 50 mM MgCl$_2$, 0.1% Tween 20, 1 mM Levamisole) after washing.

For brain imaging, confocal Z stacks were taken using a Leica SP5/SP2, using a 40x-oil objective. For the quantification of stained Kenyon cells, the cross-section equal to the width of the calyx was scanned and the fluorescence intensity quantification was performed as previously described using ImageJ[24]. For the imaging of the calyces of the mushroom bodies of honeybee brains, single optical sections were taken in the *x-y* plane. The image acquisition settings were kept identical for all preparations.

**Protein modeling and sequence analysis**. Structural modelling of *Apis melifera* ELAVL2 splice variants was performed in SWISS-MODEL[90]. For RRM1/2 modelling, the HuR RRM1/2 structure was used as a template (PDB accession:

4ed5.1.A). For RRM3 modelling, HuR RRM3 structure was used as a template (PDB accession: 6gd3.1). The hinge region could not be modelled due to a lack of known structures with a sufficient degree of homology. Structural features in the hinge region between RRMS2 and 3, and the loop consisting of variable exons 3a and 4a inserted between beat sheets 2 and 3 in RRM2 were predicted using the JPRED webserver[91]. Predicted secondary structure of alpha-helices in alternative exons 3a and 4a were manually added to the model.

The sequence of alternatively spliced exon 4c was retrieved either from annotations for splice variants from humans[80], a cricket[19], silk moth and a mosquito[18], by analysing sequences annotated in the UCSC genome browser (www.genome.ucsc.edu) for a chicken, *Xenopus* and zebrafish, or from annotations in GenBank (https://www.ncbi.nlm.nih.gov/gene/) for a lancelet (LOC118431358) and lamprey (LOC116951932) and sea urchin (LOC115928867) or for a sea squirt in Ensembl (www.ensembl.org, ENSCSAVG00000003440).

In addition, we validated the annotation of the sea urchin *Strongylocentrotus purpuratus* exon 4c by retrieving sequence reads from expression data from deeply sequenced transcriptomes[92] by realigning RNA sequencing reads using STAR[93] (version 2.7.2; parameters: alignSJDBoverhangMin 3 —twopassMode Basic — alignIntronMin 5 —alignMatesGapMax 200000 —alignIntronMax 200000. The genome index has been generated using the *S. purpuratus* genome version 5 as downloaded 25th of July 2021 from Echinobase[94]. The index was built without primary gene annotation. The results have been evaluated against the genome annotation using IGV and in-house scripts to assess the expression level of the exon. Of 23 samples, two showed clear expression of the exon.

**Statistics and reproducibility**. Multiple planned pairwise comparisons of expression levels were done by ANOVA followed by Fisher's protected least significance difference post hoc test using StatView. To compare proportions of conditioned responses between groups, a repeated-measure analysis of variance (ANOVA) was run for the acquisition data (one factor, *treatment*, with *trial* as the repeated measure), and a simple ANOVA for retention data[95]. Post hoc comparisons of rates of CS-specific responses were done using the Fisher's exact test.

**Reporting Summary**. Further information on research design is available in the Nature Research Reporting Summary linked to this article.

## Data availability
All data are available in the main text or the supplementary material (Supplementary Data 1). Reagents are available upon reasonable request from the corresponding author.

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

## Acknowledgements

We thank the Winterbourne garden (Birmingham) and Lucie Hotier (Toulouse) for providing bees, Valérie Hilgers for rabbit anti-ELAV antibodies, Karthik Nallasivan for help with imaging, Reinhard Stöger for discussions and comments on the manuscript. For this work we acknowledge funding from the Sukran Sinan Fund, the Genetics Society, the Biochemical Society and BBSRC. JMD acknowledges funding from the CNRS and Université Paul Sabatier.

## Author contributions

P.U., I.U.H., T.D. and M.S. performed molecular biology experiments, P.U., J.K.G., and N.D. performed behavioral experiments, and J.K.G., P.U. and T.D. performed antibody stainings and in situ hybridization experiments and imaging. J.M.D. designed behavioral experiments. T.D. analyzed structures and A.R. provided bioinformatics support for sequence analysis. MS conceived the project and wrote the original draft of the manuscript. J.M.D., I.U.H, and all other authors reviewed and edited. M.S. and J.M.D. supervised and acquired funding.

## Competing interests

The authors declare no competing interests.
