## [Peer Review File · Communications Biology]

Reviewers' comments:

Reviewer #1 (Remarks to the Author):

To the Authors

The main objective of the paper was investigating the consolidation of the memory using a known model, the ELAV, a highly conserved RNA binding protein that acts in the CNS regulating polyadenylation sites in neurons, determining the extended of 3'UTR landscape (briefly from Wei, et al., 2020).

Abstract

Please comment:

The micro exon (HU) has already been discussed by Decio et al (2019) as 4c, the same authors also referred to *Apis mellifera*'s ELAV / HU as a single gene. (Correctly cited at Line 87)

Suggestion: I am requesting a better presentation and discussion of this item throughout the text.

Introduction

L 93: Change individual to individual.

L 149-150: ... Exon 4d is only 3 nt long (see also L 173)

Confirm 3 nt long, I found references to 4 nt.

L 173: Bees also have a 3 nt micro exon

My concern: Carrasco et al., (2021, 2020) refer to a 45 nt mini exon, also conserved between invertebrates (*Drosophila*) and humans. I am asking: is it a second exon or the same? Clarify. I found a paper that refers to a micro exon, however from 1998. (Wei et al 2020 also refer to 4c "micro exon". It would be useful to clarify this information).

L 191 on....:

Question/ Concern: It is not clear if was observed a differential use of the alternative forms of ELAV, when worker's brains were analyzed.

L 207: Consistent with its nuclear localization... in about 75% of the worker bee brains...

Concern: Then, it is not preferentially expressed in specific parts of the brain.

Clarify.

L 209 – 231

In this part of the paper, I have my main concerns and questions.

1) L 209: Are the histological preparations clear enough? I made some observations in the figures.

2) L 211 – 216: ... ELAV expression might be indicative

All bees are supposed to have the same duties and experience. Why the age and behavior of the bees were not controlled?

3) L 216in some of the brains ...

In the hive it is possible to observe different worker bees, according to age and stage of development: nurse (young bees) in different stages of development, foragers (older) performing different functions, and some workers called generalists. These have both functions in a normal colony. What do the authors think about this possibility? In addition, some bees will never be foragers.

Discussion

L 286: Is had had correct? Verify the sentence.

L 287:

I am suggesting explain better what "molecular tools currently lacking in the honey bees..." means.

Material and Methods

L 403 on

Consider rewriting the paragraph or building a table for the primers.

(I understand that this methodology is classic, however, to be clearer it could be broken down according to the protocols or the identified primers could compose a table).

Figures 4 and 5:

Consider identifying parts of the brain to aid the reader's understanding.

Reviewer #2 (Remarks to the Author):

The manuscript from Ustaoglu et al., entitled "Dynamically expressed ELAV is required for learning and memory in bees" deals with the expression analysis of a single Hu/elav-family gene in the honey bee system to access the possible function of the gene in learning and memory. In general, it is believed that the Hu/elav-family genes play roles in the differentiation and maintenance of the neurons. In this paper, the authors reported that a Hu/elav homolog in the honey bee plays a role in neuronal plasticity because RNAi-knockdown of the Hu/elav homolog impaired memory formation. Besides, they found that the expression levels of the splice variant of *Apis* Hu/elav homolog change during memory consolidation. It is fascinating if their conclusions are TRUE. But I found too many severe concerns for the logic and the designs of the study. Besides, I found some ethical issues in the manuscript. I could not judge whether this study is worth publishing unless the authors appropriately answer all of the following questions/concerns.

Major comments:

1. The identification, expression analysis, and prediction of protein 3D conformation of the splice variants of the insect Hu/elav homolog have been already reported by Watanabe and Aonuma (2013). The paper was one of the first studies on the insect Hu/elav-family genes in non-Dipteran insects. Sadly, the authors completely ignored the previous reports; even their experimental design of the molecular biology part generally followed Watanabe's previous work. The authors MUST cite Watanabe's previous work and compare their results to the previous findings in the cricket appropriately.

2. As the authors mentioned in the Introduction section, *Drosophila* possesses three Hu/elav-family genes, *elav*, *rbp9*, and *fne*. The honeybee has only one Hu/elav-like gene that structurally resembles the *Drosophila fne*. The name of the Hu/elav-like gene in *Apis* should be *found in neurons* (*fne*), not *elav*. The phylogenetic analysis conducted by Watanabe and Aonuma (2013) also supports this idea.

*Hereafter, to avoid confusion, the Hu/elav-family gene of the honey bee is referred to as "*Apis fne*."

3. Is *Apis fne* specifically expressed in the nervous system of the honey bee? Watanabe and Aonuma (2013) reported that the *fne* homolog in the cricket is broadly expressed in various tissues. Suppose *Apis fne* shows a similar broad expression pattern in the honeybee tissues as the case in the cricket. In that case, it becomes difficult to discuss the underlying mechanism of RNAi-induced memory deficiency phenotype.

4. The most critical concern of this study is the specificity of the anti-Elav antibody. The authors used the rat monoclonal antibody raised against the recombinant full-length protein of *Drosophila* Elav (rat anti-DmElav mAb). *Drosophila* possesses three Hu/elav-family genes, and they are all expressed in the nervous tissue, and the rat anti-elav mAb was proved to specifically immunoreact with the *Drosophila* Elav. Therefore, it is evident that the epitope of anti-DmElav mAb is specific in *Drosophila* Elav, and *Drosophila* Rbp9 and Fne might not contain the epitope.

Then, here is the question: Does the rat anti-DmElav mAb immunoreact with the *Apis* Fne?

Drosophila Fne protein shows the highest sequence similarity to *Apis* Fne among three *Drosophila* Hu/elav-like proteins. As shown in Fig. S1, *Drosophila* Elav and *Apis* Fne shared three RRM domains, but the linker regions in the proteins are not conserved. Therefore, if the rat anti-DmElav mAb immunoreacts with *Apis* Fne, the epitope of the mAb should locate within the conserved RRM domains. And such antibodies can immunoreact with the other Hu/elav-like proteins in *Drosophila*. The authors need to solve this contradiction!

The assessment of the RNAi efficiency and the intracellular protein localization entirely relies on the anti-DmElav mAb. Therefore, the specificity of the anti-DmElav mAb to *Apis* Fne protein must be ensured.

Here are the specific questions/concerns about the antibodies:

Did the size of the immunoreactive band match the calculated molecular mass of *Apis* Fne protein isoforms?

The author MUST show the full picture of the immunoblot with an appropriate size marker.

The author detected various splice variants of *Apis* Fne in the brain. Then, why the immunoblot of *Apis* Fne appeared as a single band?

The original paper of the rat anti-DmElav mAb is Rubin (1994), not Yannoni et al. (1999).

Did the authors obtain the rat anti-DmElav mAb from DSHB? Please describe the detail of the antibody.

The information about the anti-Tub antibody is missing in the manuscript.

Describe the detail on the quantification of the *Apis* Fne protein expression level in the Materials and Methods section. Soller and White (2005) did not measure the protein expression level with Western blotting.

5. The next big concern of this study is *in situ* hybridization. The authors conducted whole-mount *in situ* hybridization according to the method described in Haussmann et al. (2008). Sadly, Haussmann et al. (2008) did not provide detailed information on the experimental procedure. The authors are required to provide sufficient information in the manuscript, or they should cite appropriate original papers to ensure reproducibility. Besides, Haussmann et al. (2008) conducted *in situ* hybridization in *Drosophila* embryos. The volume of the honeybee brain is much larger than that of *Drosophila* embryos. We need more details to judge whether the staining data are reliable or not. Overall, the experimental procedures provided in the current version of the manuscript were insufficient.

Here are some questions/concerns:

Why did not the authors adjust the concentration of DIG-labeled probes in each experiment?

Many researchers conduct *in situ* hybridization on the frozen honeybee brain sections. Did the authors performed *in situ* hybridization on frozen sections to ensure their results with the whole-mount preparations?

Did the authors set negative controls using the sense probes or genes not encoded in the honey bee genome (e.g., beta-lactamase or GFP)? Without negative control experiments, it is impossible to interpret the results of *in situ* hybridization.

6. The Introduction section of the manuscript begins with the topic of immediate-early genes (IEGs), and the authors listed 'IEG' in the Keyword of the paper. Immediate-early genes (IEGs) are the genes whose transcription is activated in response to stimuli. The mRNAs of IEGs are rapidly and transiently induced, and their expression does not require *de novo* protein synthesis. The author provided no evidence that *Apis* fne is expressed as an IEG or *Apis* Fne regulates the splicing of IEG products. Therefore, it is apparent that this study does not deal with IEGs in the honeybee nervous system. Statements about IEG need to be deleted from the manuscript.

7. I can't entirely agree with the idea that the genetic diversity of organisms directly links to the complexity of the life history, which the authors mentioned between lines 139-143.

8. There is a concern about self-plagiarism. Dynamic regulation of the splicing of *Apis* fne gene was already reported in Decio et al. (2019), a Scientific report paper from the same research group. Sadly, the authors did not cite their previous report appropriately in the Result section, and to make matters worse, they re-used one figure (Figure 5A in Decio et al., 2019 -> Figure 2A of this manuscript).

9. The deduced amino acid sequence of the microexon 4c and those of its corresponding exons in vertebrate Hu/elav-family genes are not conserved (Fig 2D). Therefore, it is difficult to assume that the microexon is evolutionally ancient based on the sequence conservation of the microexon. Besides, it is not surprising that the insect *fne* gene has an ancestral gene organization. Samson (2008) had already proposed the evolutionary history of the insect Hu/elav-family genes: the *fne* gene is the most ancestral Hu/elav-family gene in insects. Dipteran insect acquired the *rpb9* gene by gene duplication of the *fne* gene, and the intron-less *elav* gene is retro-transposed in origin. The phylogenetic analysis of the insect Hu/elav-family genes conducted by Watanabe and Aonuma (2013) supports this idea. And

Watanabe and Aonuma have already reported existence of various splice variants of the *fne* homolog in the cricket.

10. The concern about the RNAi experiment: Considering the doubt on the antibody specificity, the decrease in the expression of the *Apis fne* gene must be double-checked by RT-qPCR. Besides, the authors should conduct time-course expression analyses after the injection of dsRNA to ensure dsRNA injection and behavioral experiments were conducted at the appropriate timing.

11. The authors conducted the phylogenetic analysis with four sequences (Fig. S1B). The number of the sequences is too less! They need to add more sequences to draw a more informative tree. Besides, Watanabe and Aonuma (2013) already reported the phylogenetic tree of the insect *Hu/elav*-family genes with a larger dataset, including the predicted *Apis fne*.

12. In Fig. S2, some 3D models of the RRM domains are connected with handwritten lines because the corresponding inserted regions are 'unstructured.' Did the authors confirm that the inserted regions in the RRM domain and the hinge region of *Apis Fne* variants are unstructured? They should at least conduct a prediction of the secondary structure of these regions, and they should properly render 3D models without handwriting.

Minor comments:

Although I noticed typos and the misuse of technical terms etc., throughout the manuscript, the authors need to answer the major comments. I will wait for the revised manuscript.

Reviewer #3 (Remarks to the Author):

Dr. Ustaoglu and his colleagues investigated how the processes of memory and learning can be modulated by the expression of variant transcripts generated by mRNA alternative splicing. However, the authors said that "little is known" about this issue, but at least a hundred articles related to this topic, containing investigations in different organisms, are easily found in searches carried out on PubMed and Scholar Google by associating keywords such as learning/memory/alternative splicing.

It should be noted that there are at least two previous studies in the same direction carried out with the bee species *Apis mellifera* (the same biological model used by Ustaoglu et al.), and that could be considered by the authors in both sections, Introduction and Discussion, of the manuscript:

Expression and localization of glutamate-gated chloride channel variants in honeybee brain (*Apis mellifera*)

<https://doi.org/10.1016/j.ibmb.2012.10.003>

Differential involvement of glutamate-gated chloride channel splice variants in the olfactory memory processes of the honeybee *Apis mellifera*

<https://doi.org/10.1016/j.pbb.2014.05.025>

The authors focused on the *elav* gene that encodes neuronal-biased RNA binding proteins and on the relationship of their expressed variants/isoforms. As their outstanding results, they concluded that the *elav* gene is alternatively spliced and the ELAV protein is required for learning and memory consolidation. In this sense, the study is original and probably will influence thinking in the field, as well as it will immediately appeal to a broad audience of readers in the scientific community. I suggest that the authors standardize the use of the term variant(s) for mRNA(s) and isoform(s) for protein(s).

The study is really well done and the text is well written. On the other hand, authors often use words whose connotation has the sense of exaggeration (such as: fundamental, pivotal, highly, most, mostly, prominent, prominently, and so on). Many of these words fit the context, but my suggestion is to revise the manuscript trying to avoid them whenever possible. Because it is a

dense and complex research, the text is long and quite technical. I suggest a re-reading of the manuscript and adjustments to the wording to make the next version more attractive and clear.

In general, the Methodology section needs to provide more details both to ensure the readers' complete understanding and the possibility that the study can be reproduced by any researcher and laboratory in the world. It is important to inform the GeneID and access numbers (XM_) of the variants analyzed based on the GenBank-NCBI, as well as the information and the ELAV motifs/domains codes already described (such as cl36948). Statistical analyzes seem appropriate for me (as I see them in the results descriptions), but they are not described in the Methodology. Moreover, please check if the p-value = 0.13 informed in line 225 and Figure 6B is correct. If correct, which statistical test was used?

The quality of the data is excellent, but the general quality of the presentation (mainly Figure 2 [A, B, C and D], Figure 3 [G, H and I] and Suppl. Figure 1) could be improved.

Please below our responses to reviews comments

Reviewer #1

We thank this reviewer for the constructive comments and spotting the error in nomenclature of alternatively spliced exons in bee ELAVL2, which we have corrected as detailed below.

Abstract

Please comment:

The micro exon (HU) has already been discussed by Decio et al (2019) as 4c, the same authors also referred to *Apis mellifera*'s ELAV / HU as a single gene. (Correctly cited at Line 87)

Suggestion: I am requesting a better presentation and discussion of this item throughout the text.

We apologize for this fundamental error in nomenclature in Decio et al, 2019. The correct nomenclature is as in the current manuscript (4c here is 4d in Decio et al.). Accordingly, we have submitted a correction for Decio et al (2019).

Introduction

L 93: Change individual to individual.

We have amended the text accordingly.

L 149-150: ... Exon 4d is only 3 nt long (see also L 173)

Confirm 3 nt long, I found references to 4 nt.

We confirm that exon 4d is 3 nts long and have checked the MS for consistency.

L 173: Bees also have a 3 nt micro exon

My concern: Carrasco et al., (2021, 2020) refer to a 45 nt mini exon, also conserved between invertebrates (*Drosophila*) and humans. I am asking: is it a second exon or the same? Clarify. I found a paper that refers to a micro exon, however from 1998. (Wei et al 2020 also refer to 4c "micro exon". It would be useful to clarify this information).

The 45 nt minixon in *fne* from Carrasco et al 2020 and the 4c microexon from Wei et al, 2020 is the same as the microexon 4c in our MS. We have added a sentence to the MS to clarify this issue. Also, we have added the reference Inman et al 1998 and Okano and Darnell 1997, first describing the alternative splicing in the hinge region of ELAVL4.

L 191 on.....:

Question/ Concern: It is not clear if was observed a differential use of the alternative forms of ELAV, when worker's brains were analyzed.

We have added a statistical analysis of splicing differences to Figs 3G-H and amended the text accordingly to detail which changes are significant between different developmental stages.

L 207: Consistent with its nuclear localization... in about 75% of the worker bee brains...

Concern: Then, it is not preferentially expressed in specific parts of the brain.

Clarify.

We have amended this part to better describe expression of ELAV in bee brains and mushroom bodies. We also included an image of a bee brain in Suppl Fig 5 annotating the different brain parts and an anti-ELAV staining of an entire bee brain.

L 209 – 231

In this part of the paper, I have my main concerns and questions.

1) L 209: Are the histological preparations clear enough? I made some observations in the figures.

We now have included explanations to the text detailing the histological observations.

2) L 211 – 216: ... ELAV expression might be indicative

All bees are supposed to have the same duties and experience. Why the age and behavior of the bees were not controlled?

We have used worker bees, which are between 5-15 d old. These bees are foragers that collect pollen and nectar and thus have similar duties and experiences.

3) L 216in some of the brains ...

In the hive it is possible to observe different worker bees, according to age and stage of development: nurse (young bees) in different stages of development, foragers (older) performing different functions,

and some workers called generalists. These have both functions in a normal colony. What do the authors think about this possibility? In addition, some bees will never be foragers. Indeed, these are very interesting avenues to follow up for future publications. However, to analyse such bees in detail, we would need to substantially upgrade our apiary to collect bees at different stages.

Discussion

L 286: Is had had correct? Verify the sentence.

This is now corrected to "have had".

L 287: I am suggesting explain better what "molecular tools currently lacking in the honey bees..." means.

We explained now which tools are missing in honey bees and added a reference.

Material and Methods

L 403 on

Consider rewriting the paragraph or building a table for the primers.

(I understand that this methodology is classic, however, to be clearer it could be broken down according to the protocols or the identified primers could compose a table).

We have generated a table to show the primer sequences, and also the sequences of probes used for in situ hybridizations. In addition, we have broken down the text to be more accessible to the reader.

Figures 4 and 5:

Consider identifying parts of the brain to aid the reader's understanding.

We have added a picture of a bee brain and labelled individual anatomical structures to Suppl Fig 5.

Reviewer #2 (Remarks to the Author):

The manuscript from Ustaoglu et al., entitled "Dynamically expressed ELAV is required for learning and memory in bees" deals with the expression analysis of a single Hu/elav-family gene in the honey bee system to access the possible function of the gene in learning and memory. In general, it is believed that the Hu/elav-family genes play roles in the differentiation and maintenance of the neurons. In this paper, the authors reported that a Hu/elav homolog in the honey bee plays a role in neuronal plasticity because RNAi-knockdown of the Hu/elav homolog impaired memory formation. Besides, they found that the expression levels of the splice variant of Apis Hu/elav homolog change during memory consolidation. It is fascinating if their conclusions are TRUE. But I found too many severe concerns for the logic and the designs of the study. Besides, I found some ethical issues in the manuscript. I could not judge whether this study is worth publishing unless the authors appropriately answer all of the following questions/concerns.

We thank this reviewer for their critical evaluation of our work and appreciate that our findings and conclusions are recognized as fascinating. As detailed below, we hope to have addressed all concerns thoroughly and appropriately to allow for publishing our exciting findings.

Major comments:

1. The identification, expression analysis, and prediction of protein 3D conformation of the splice variants of the insect Hu/elav homolog have been already reported by Watanabe and Aonuma (2013). The paper was one of the first studies on the insect Hu/elav-family genes in non-Dipteran insects. Sadly, the authors completely ignored the previous reports; even their experimental design of the molecular biology part generally followed Watanabe's previous work. The authors MUST cite Watanabe's previous work and compare their results to the previous findings in the cricket appropriately.

We deeply apologize for this omission and have now cited this highly-relevant paper by Watanabe and Aonuma (2014). In particular, we have added a paragraph to the introduction detailing in more detail the complexity of the ELAV/Hu family founded to a large degree on the work of Watanabe and Aonuma who also discovered extensive alternative splicing in the cricket ELAV/Hu orthologue found in neurons. Also, we have compared their work to our observations in the discussion section. Moreover, we apologize for unintentionally following the structure of the paper by Watanabe and

Aonuma after describing the learning phenotype, but we admit, there are not many other ways to describe this type of molecular biology work flow.

2. As the authors mentioned in the Introduction section, *Drosophila* possesses three Hu/elav-family genes, *elav*, *rbp9*, and *fne*. The honeybee has only one Hu/elav-like gene that structurally resembles the *Drosophila fne*. The name of the Hu/elav-like gene in *Apis* should be found in neurons (*fne*), not *elav*. The phylogenetic analysis conducted by Watanabe and Aonuma (2013) also supports this idea.

*Hereafter, to avoid confusion, the Hu/elav-family gene of the honey bee is referred to as "Apis *fne*." Bee *elav* is annotated as *elav-like protein 2* or *elavl2* in NCBI GenBank. Accordingly, we have changed the name to *elavl2* throughout the text.

3. Is *Apis fne* specifically expressed in the nervous system of the honey bee? Watanabe and Aonuma (2013) reported that the *fne* homolog in the cricket is broadly expressed in various tissues. Suppose *Apis fne* shows a similar broad expression pattern in the honeybee tissues as the case in the cricket. In that case, it becomes difficult to discuss the underlying mechanism of RNAi-induced memory deficiency phenotype.

We have analysed bee ELAVL2 expression by Western-blot in different tissues of bees. These results show that ELAVL2 in bees is strongly expressed in the brain but not in other major tissues. We have validated this finding with two anti-sera cross-reactive with *Apis ELAVL2*. Our previous polyclonal antisera made in rats (Robinow and White, 1991) and a recently made antiserum raised in rabbits (Carrasco et al, 2020).

However, we note that in *Drosophila elav* mRNA is more broadly expressed, which has also been found in the cricket by Watanabe and Aonuma. We have added this point to the discussion.

Also, the RNAi we did specifically in the brain by injecting the dsRNA into the mushroom body region of the central brain and observed a very specific memory defect.

4. The most critical concern of this study is the specificity of the anti-Elav antibody. The authors used the rat monoclonal antibody raised against the recombinant full-length protein of *Drosophila Elav* (rat anti-DmElav mAb).

This is fundamental mis-conception of the reviewer as we used a POLYCLONAL anti-ELAV serum that was made in Kalpana White's lab by immunizing rats and was published (Robinow and White, 1991). We now clearly state this in the text and materials and methods this fact and added the correct reference.

Moreover, we have added data showing that this POLYCLONAL anti-ELAV serum cross-reacts with bacterially expressed ELAV/Hu family proteins.

Drosophila possesses three Hu/elav-family genes, and they are all expressed in the nervous tissue, and the rat anti-elav mAb was proved to specifically immunoreact with the *Drosophila Elav*. Therefore, it is evident that the epitope of anti-DmElav mAb is specific in *Drosophila Elav*, and *Drosophila Rbp9* and *Fne* might not contain the epitope.

Then, here is the question: Does the rat anti-DmElav mAb immunoreact with the *Apis Fne*?

To specifically validate that this antibody cross-reacts with bee ELAVL2, we expressed recombinant ELAV/Hu family proteins in *E. coli* and added this data to the supplement.

Drosophila Fne protein shows the highest sequence similarity to *Apis Fne* among three *Drosophila* Hu/elav-like proteins. As shown in Fig. S1, *Drosophila Elav* and *Apis Fne* shared three RRM domains, but the linker regions in the proteins are not conserved. Therefore, if the rat anti-DmElav mAb immunoreacts with *Apis Fne*, the epitope of the mAb should locate within the conserved RRM domains. And such antibodies can immunoreact with the other Hu/elav-like proteins in *Drosophila*. The authors need to solve this contradiction!

As detailed above, we used this polyclonal anti-ELAV serum to show cross reactivity to bacterially expressed ELAV/Hu proteins. Intriguingly, this polyclonal anti-ELAV serum shows little cross-reactivity with *Drosophila FNE* and *RBP9* as shown the supplemental Fig 1.

The assessment of the RNAi efficiency and the intracellular protein localization entirely relies on the anti-DmElav mAb. Therefore, the specificity of the anti-DmElav mAb to *Apis Fne* protein must be ensured.

Apis ELAVL2 levels are reduced by 80% after RNAi. We consider this result as a very strong argument that the polyclonal anti-ELAV serum is specific to Apis ELAVL2. In addition, Western-blots with our polyclonal antibody recognizes one broad band of the predicted size of Apis ELAVL2. Moreover, we added data to Supp Fig 1 showing gradual reduction of Apis ELAVL2 levels different times after RNAi (injection of dsRNA against Apis ELAVL2 into the brain) induction.

Here are the specific questions/concerns about the antibodies:

Did the size of the immunoreactive band match the calculated molecular mass of Apis Fne protein isoforms?

As indicated in Suppl Fig 1C, the size observed on Western-blots matches the calculated size. In addition, the Western blots show several, close-migrating bands of the expected sizes.

The author MUST show the full picture of the immunoblot with an appropriate size marker.

We have added uncropped picture of the Western blot to Suppl Fig 1 alongside with the size markers.

The author detected various splice variants of Apis Fne in the brain. Then, why the immunoblot of Apis Fne appeared as a single band?

The difference between the different protein isoforms of Apis ELAVL2 is small, but several, close-migrating bands of the expected size are detected. We now also used infrared dye labelled secondary antibodies, which give a higher resolution than chemiluminescence imaging of Westerns.

The original paper of the rat anti-DmElav mAb is Rubin (1994), not Yannoni et al. (1999).

We have added the correct reference now describing the POLYCLONAL anti-ELAV serum, which is Rabinow and White 1991.

Did the authors obtain the rat anti-DmElav mAb from DSHB? Please describe the detail of the antibody.

We have described the details of this POLYCLONAL anti-ELAV serum published by Rabinow and White 1991 in the Methods section in detail.

The information about the anti-Tub antibody is missing in the manuscript.

We have added the information about the anti-tubulin antibody to the Methods section.

Describe the detail on the quantification of the Apis Fne protein expression level in the Materials and Methods section. Soller and White (2005) did not measure the protein expression level with Western blotting.

We now describe in the Methods section how Western blots were quantified.

5. The next big concern of this study is in situ hybridization. The authors conducted whole-mount in situ hybridization according to the method described in Hausmann et al. (2008). Sadly, Hausmann et al. (2008) did not provide detailed information on the experimental procedure. The authors are required to provide sufficient information in the manuscript, or they should cite appropriate original papers to ensure reproducibility. Besides, Hausmann et al. (2008) conducted in situ hybridization in *Drosophila* embryos. The volume of the honeybee brain is much larger than that of *Drosophila* embryos. We need more details to judge whether the staining data are reliable or not. Overall, the experimental procedures provided in the current version of the manuscript were insufficient.

We have now added a detailed description of how RNA insitu hybridizations were done to the methods section.

Here are some questions/concerns:

Why did not the authors adjust the concentration of DIG-labeled probes in each experiment?

Probes were initially tested with a NBT/BCIP AP color staining and this procedure was used to adjust the probe concentration. In addition, we include an overnight wash in 50% formamide buffer to wash off any unhybridized probe. We have added these details to the Methods section.

Also, we have added data to the supplement showing control hybridizations including a *Drosophila* probe in bee brains showing lack of staining, a Dscam probe in bee brains to show ubiquitous expression, hybridizations to *Drosophila* tissue mixed with bee tissues with the same probes.

Many researchers conduct in situ hybridization on the frozen honeybee brain sections. Did the

authors performed in situ hybridization on frozen sections to ensure their results with the whole-mount preparations?

Whole-mount preparations allow to visualize the staining in the entire mushroom body in 3D by confocal microscopy. Obtaining the same information by sectioning brains is very difficult if not impossible.

Did the authors set negative controls using the sense probes or genes not encoded in the honey bee genome (e.g., beta-lactamase or GFP)? Without negative control experiments, it is impossible to interpret the results of in situ hybridization.

We have added control stainings to the supplement detecting the *Dscam* common region to show ubiquitous expression and as a negative control used a *Drosophila tango13* probe in bee brains to show absence of staining.

Moreover, we added data to the supplement from hybridizing a species-specific probe to *Drosophila* embryos and bee brains in the same tube to demonstrate species-specificity probes under the same hybridization conditions to show that *Drosophila elav* only hybridizes to *Drosophila* but not bee mushroom bodies, and that bee exon 4c hybridizes to bee mushroom bodies but not to *Drosophila* embryos.

6. The Introduction section of the manuscript begins with the topic of immediate-early genes (IEGs), and the authors listed 'IEG' in the Keyword of the paper. Immediate-early genes (IEGs) are the genes whose transcription is activated in response to stimuli. The mRNAs of IEGs are rapidly and transiently induced, and their expression does not require de novo protein synthesis. The author provided no evidence that *Apis fne* is expressed as an IEG or *Apis Fne* regulates the splicing of IEG products. Therefore, it is apparent that this study does not deal with IEGs in the honeybee nervous system. Statements about IEG need to be deleted from the manuscript.

We have deleted this part about immediate-early genes from the introduction and explained better in the text, that an early phase of memory consolidation requires transcription without protein synthesis. We now specify that the changes occurring in ELAV expression and alternative splicing occur in the same time-frame as the induction of IEGs in the discussion section.

7. I can't entirely agree with the idea that the genetic diversity of organisms directly links to the complexity of the life history, which the authors mentioned between lines 139-143.

This is a mis-conception from this reviewer as genic, not genetic diversity is meant. We have rephrased this sentence for clarity.

8. There is a concern about self-plagiarism. Dynamic regulation of the splicing of *Apis fne* gene was already reported in Decio et al. (2019), a Scientific report paper from the same research group. Sadly, the authors did not cite their previous report appropriately in the Result section, and to make matters worse, they re-used one figure (Figure 5A in Decio et al., 2019 -> Figure 2A of this manuscript).

We disagree with this reviewer that gene models can be plagiarised. In fact, using the same graphics helps to avoid confusion. We have now cited Decio et al 2019 for initial characterization of splicing in *apis elav/2* and altered the gene model such that it is not an exact duplication.

9. The deduced amino acid sequence of the microexon 4c and those of its corresponding exons in vertebrate Hu/*elav*-family genes are not conserved (Fig 2D). Therefore, it is difficult to assume that the microexon is evolutionally ancient based on the sequence conservation of the microexon.

We have added more species between insects and vertebrates to strengthen the functional significance of microexon 4c. Moreover, we have added a consensus motif for exon 4c to Fig 2D. Microexon 4c is found at the same position in the unstructured and very diverse hinge region. This exon is identical between jawless fish and the three human Hu proteins, which is very unexpected for an unstructured part of a protein that rapidly diverges around this exon 4c. For a more balanced approach we have added to the discussion the possibility of convergent evolution.

Besides, it is not surprising that the insect *fne* gene has an ancestral gene organization. Samson (2008) had already proposed the evolutionary history of the insect Hu/*elav*-family genes: the *fne* gene is the most ancestral Hu/*elav*-family gene in insects. Dipteran insect acquired the *rbp9* gene by gene duplication of the *fne* gene, and the intron-less *elav* gene is retro-transposed in origin. The phylogenetic analysis of the insect Hu/*elav*-family genes conducted by Watanabe and Aonuma (2013) supports this idea. And Watanabe and aonuma have already reported exsistance of various splice variants of the *fne* homolog in the cricket.

ELAV family genes have a conserved gene structure and microexon 4c is found at the same position in the unstructured and very diverse hinge region. That this exon arose in vertebrate and insect ancestral genes independently is thus very unlikely, but for a more balanced approach we have added to the discussion the possibility of convergent evolution. We have cited Watanabe and Aonuma for describing alternative splicing in this region of cricket FNE.

10. The concern about the RNAi experiment: Considering the doubt on the antibody specificity, the decrease in the expression of the *Apis fne* gene must be double-checked by RT-qPCR. As detailed above we have clarified the polyclonal anti-ELAV serum is specific to *Apis ELAVL2* and directly determined reduction of protein expression after RNAi.

Besides, the authors should conduct time-course expression analyses after the injection of dsRNA to ensure dsRNA injection and behavioral experiments were conducted at the appropriate timing. We have done time course-experiments to validate that after 2 days RNAi is effective and added this data to Suppl Fig 1.

11. The authors conducted the phylogenetic analysis with four sequences (Fig. S1B). The number of the sequences is too less! They need to add more sequences to draw a more informative tree. Besides, Watanabe and Aonuma (2013) already reported the phylogenetic tree of the insect *Hu/elav*-family genes with a larger dataset, including the predicted *Apis fne*.

We agree, that more complete phylogenetic trees have been published. We refer to them in the text now and have removed the phylogenetic tree. Showing the sequence alignment here, however, is important to indicate where the alternatively spliced parts are.

12. In Fig. S2, some 3D models of the RRM domains are connected with handwritten lines because the corresponding inserted regions are 'unstructured.' Did the authors confirm that the inserted regions in the RRM domain and the hinge region of *Apis Fne* variants are unstructured? They should at least conduct a prediction of the secondary structure of these regions, and they should properly render 3D models without handwriting.

We have redone the structural analysis shown in Supplemental Fig 3 to analyse the unstructured regions. For the 3a/4a region this could be modelled, while for the hinge region we included a separate analysis. In addition, we have properly rendered the models without handwriting.

Minor comments:

Although I noticed typos and the misuse of technical terms etc., throughout the manuscript, the authors need to answer the major comments. I will wait for the revised manuscript.

We have checked the MS again for typos and misuse of technical terms.

Reviewer #3 (Remarks to the Author):

We thank this reviewer for their appreciation of our work in being influential to the field, their constructive comments and highlighting highly relevant previously published studies to cite.

Dr. Ustaoglu and his colleagues investigated how the processes of memory and learning can be modulated by the expression of variant transcripts generated by mRNA alternative splicing. However, the authors said that "little is known" about this issue, but at least a hundred articles related to this topic, containing investigations in different organisms, are easily found in searches carried out on PubMed and Scholar Google by associating keywords such as learning/memory/alternative splicing.

It should be noted that there are at least two previous studies in the same direction carried out with the bee species *Apis mellifera* (the same biological model used by Ustaoglu et al.), and that could be considered by the authors in both sections, Introduction and Discussion, of the manuscript:

Expression and localization of glutamate-gated chloride channel variants in honeybee brain (*Apis mellifera*)

<https://doi.org/10.1016/j.ibmb.2012.10.003>

Differential involvement of glutamate-gated chloride channel splice variants in the olfactory memory processes of the honeybee *Apis mellifera*

<https://doi.org/10.1016/j.pbb.2014.05.025>

We have addressed this issue by expanding the introduction accordingly and cited this highly relevant work and others. In particular, we point out that alternative splicing in an number of genes has been linked to learning and memory, but how RNA binding proteins coordinate alternative splicing in learning in memory has not been explored.

The authors focused on the elav gene that encodes neuronal-biased RNA binding proteins and on the relationship of their expressed variants/isoforms. As their outstanding results, they concluded that the elav gene is alternatively spliced and the ELAV protein is required for learning and memory consolidation. In this sense, the study is original and probably will influence thinking in the field, as well as it will immediately appeal to a broad audience of readers in the scientific community. I suggest that the authors standardize the use of the term variant(s) for mRNA(s) and isoform(s) for protein(s).
We have changed the nomenclature accordingly to say splice variants and protein isoforms.

The study is really well done and the text is well written. On the other hand, authors often use words whose connotation has the sense of exaggeration (such as: fundamental, pivotal, highly, most, mostly, prominent, prominently, and so on). Many of these words fit the context, but my suggestion is to revise the manuscript trying to avoid them whenever possible. Because it is a dense and complex research, the text is long and quite technical. I suggest a re-reading of the manuscript and adjustments to the wording to make the next version more attractive and clear.
We have toned down the text to remove exaggeration and reworded technical parts to be more attractive to a broader audience.

In general, the Methodology section needs to provide more details both to ensure the readers' complete understanding and the possibility that the study can be reproduced by any researcher and laboratory in the world.
To improve reproducibility we have added more detail including full protocols for in situ hybridizations. Likewise, we have removed primer sequences and add them to a separate table.

It is important to inform the GeneID and access numbers (XM_) of the variants analyzed based on the GenBank-NCBI, as well as the information and the ELAV motifs/domains codes already described (such as cl36948).
We have added accession numbers of ELAV family proteins to the legend of Supplemental Fig 2. We further detailed where we have identified exon 4c in other species in Materials and Methods. Unfortunately, Genebank does not take submissions for alternatively spliced exons, only for full length cDNAs. Since, we have not done long-read sequencing nor have we cloned Apis ELAV sequences in depth to obtain all combinations of isoforms, we have added Genbank or Ensembl accession numbers for exon 4c.

Statistical analyzes seem appropriate for me (as I see them in the results descriptions), but they are not described in the Methodology. Moreover, please check if the p-value = 0.13 informed in line 225 and Figure 6B is correct. If correct, which statistical test was used?
Statistical analysis is described in the methods section at the end. In Fig 6B, differences in gene expression levels are plotted as fold change and the inadequate p-value has been removed.

The quality of the data is excellent, but the general quality of the presentation (mainly Figure 2 [A, B, C and D], Figure 3 [G, H and I] and Suppl. Figure 1) could be improved.
We have improved the general presentation of Fig 2 now including depiction of the different combinations of exon 4, 4a, 4b and 4a,b, and simplification of the sequence presentation in Fig 2C. In Fig 3F-H we have added brackets to indicate significant differences.

REVIEWERS' COMMENTS:

Reviewer #1 (Remarks to the Author):

To Authors

Thanks to the authors for the careful review of the manuscript.

The manuscript under review is a detailed description of the role of ELAV in bees submitted to different tests related to learning and long-term memory.

The text received an intense review and met most of the reviewers' suggestions. What I noticed throughout the reading, some references are not really related to what the authors suggest. I didn't correlate them all, obviously, but I needed some information and when I read the original work I didn't find what I needed. As an example I can mention:

L 101-102: ...expression and inclusion levels of alternative exons change during the early phases of memory

102 consolidation that requires transcription, but not protein synthesis (References 38, 39).

My concern: In Leder et al it is unclear whether they are referring to the inclusion of an alternative exon, they discuss the second wave of transcription.

My concern: Thus, I suggest that the concepts that underlie the discussion be carefully reviewed.

Reviewer #2 (Remarks to the Author):

The authors revised the manuscript thoroughly (especially in the materials and methods section) and added new supplementary data. The authors answered most of the concerns raised in the initial manuscript properly. In addition, the authors properly valid ethical concerns found in the initial manuscript. I am happy with this version of manuscript and I suggest to be published.

Reviewer #3 (Remarks to the Author):

By addressing the suggestions and comments of all reviewers, the authors substantially improved the scientific quality of the manuscript, supporting its publication in the journal 'Communications Biology'.

Please below our responses to reviewer 1's minor comment

Reviewer #1

Thanks to the authors for the careful review of the manuscript.

The manuscript under review is a detailed description of the role of ELAV in bees submitted to different tests related to learning and long-term memory.

The text received an intense review and met most of the reviewers' suggestions. What I noticed throughout the reading, some references are not really related to what the authors suggest. I didn't correlate them all, obviously, but I needed some information and when I read the original work I didn't find what I needed. As an example I can mention:

L 101-102: ...expression and inclusion levels of alternative exons change during the early phases of memory consolidation that requires transcription, but not protein synthesis (References 38, 39).

My concern: In Leder et al it is unclear whether they are referring to the inclusion of an alternative exon, they discuss the second wave of transcription.

My concern: Thus, I suggest that the concepts that underlie the discussion be carefully reviewed.

In addition to protein synthesis from stored mRNA, memory consolidation also requires transcription in bees as shown in Lefer et al. Hence, it is possible that during transcription alternative splicing can change and thus impact on memory consolidation.

According to Rev1's suggestions we have reviewed the concepts of how gene expression impacts on memory consolidation in the introduction and discussion. In particular, we have added the following clarification to the end of the introduction: "In this memory consolidation phase, also transcription is required and hence alternative splicing can be altered then depending on experience^{38,39}. In addition, we have amended the discussion to say that "memory consolidation requires transcription in addition to protein synthesis from stored mRNA at synapses^{4,40,41}", and added an additional reference.